# Fungal Metabolite Asperaculane B Inhibits Malaria Infection and Transmission

**DOI:** 10.3390/molecules25133018

**Published:** 2020-07-01

**Authors:** Guodong Niu, Yue Hao, Xiaohong Wang, Jin-Ming Gao, Jun Li

**Affiliations:** 1Department of Biological Sciences, Biomolecular Sciences Institute, Florida International University, Miami, FL 33199, USA; gniu@fiu.edu (G.N.); xiawang@fiu.edu (X.W.); 2College of Public Health, South China University, Hengyang, Hunan 421001, China; yhao@fiu.edu; 3Shaanxi Key Laboratory of Natural Products Chemical Biology, College of Science, Northwest A&F University, Yangling, Shaanxi 712100, China; jinminggao@nwafu.edu.cn

**Keywords:** mosquito, malaria, transmission, FREP1, multiple functional drugs, fungal secondary metabolites

## Abstract

Mosquito-transmitted *Plasmodium* parasites cause millions of people worldwide to suffer malaria every year. Drug-resistant *Plasmodium* parasites and insecticide-resistant mosquitoes make malaria hard to control. Thus, the next generation of antimalarial drugs that inhibit malaria infection and transmission are needed. We screened our Global Fungal Extract Library (GFEL) and obtained a candidate that completely inhibited *Plasmodium falciparum* transmission to *Anopheles gambiae*. The candidate fungal strain was determined as *Aspergillus aculeatus*. The bioactive compound was purified and identified as asperaculane B. The concentration of 50% inhibition on *P. falciparum* transmission (IC_50_) is 7.89 µM. Notably, asperaculane B also inhibited the development of asexual *P. falciparum* with IC_50_ of 3 µM, and it is nontoxic to human cells. Therefore, asperaculane B is a new dual-functional antimalarial lead that has the potential to treat malaria and block malaria transmission.

## 1. Introduction

Malaria is the primary tropical disease in the world. It causes nearly 300 million clinical malaria cases and half a million deaths annually [1]. Malaria prevention and elimination programs face challenges using traditional vector control methods since resistance to the main classes of insecticides has spread to all vectors in all malaria-endemic countries [2]. Furthermore, the *Plasmodium* parasites have developed resistance against the most commonly used antimalarial drugs throughout Africa and Asia [3], especially amid concerns about the spread of artemisinin resistance over the last five years. This process may accelerate in the future [4,5]. The international malaria community urges new strategies and next-generation drugs to control malaria.

To discover new drug leads, screening structurally diverse synthetic or natural small-molecule libraries with high-throughput screening (HTS) led to identifying several novel antimalarial compounds [6]. Among these screening methods, *Plasmodium falciparum* growth assays with cultured parasites are relatively robust [7], and most of these screenings target asexual stage malaria. Transmission-blocking small molecules were rarely reported because transmission-blocking reagents need to focus either on the sexual stage of parasites or malaria vectors. In contrast, disease transmission molecular mechanisms are not well understood.

To our knowledge, HTS against FREP1-*Plasmodium* interaction is the only approach targeting malaria transmission [8]. The target protein, fibrinogen-related protein 1 (FREP1) from *Anopheles gambiae*, is critical for parasites to infect the mosquito midgut [9] through its interaction with gametocytes or ookinetes [10]. This conserved pathway exists across multiple species of *Plasmodium* and *Anopheles* [11]. Therefore, the drugs discovered by this approach are potentially functional for controlling all malaria pathogens and vectors to spread malaria.

This study screened a fungal extract library (Global Fungal Extract Library, GFEL) and obtained a candidate extract that completely blocked *P. falciparum* transmission to *An. gambiae* at 100 µg/mL. Further assays identified the fungal species as *Aspergillus aculeatus*. Many bioactive secondary metabolites have been isolated from this fungal species [12,13]. The active compound was isolated and determined to be asperaculane B. The structure of asperaculane B has resolved previously [12,14]. Notably, asperaculane B also inhibited asexual *P. falciparum* infection in blood while being nontoxic to human cells. Therefore, asperaculane B reported in this paper is a promising lead in developing a drug that treats malaria patients as well as blocks the spread of malaria.

## 2. Results

### 2.1. Fungal Extract GFEL-46E5 Significantly Inhibits P. Falciparum Transmission to An. Gambiae

Using our previously established method [8], we screened our fungal extract library with about 10,000 extracts from different fungal isolates separated from soil and plant samples collected worldwide. A fungal extract named as GFEL-46E5, representing a fungal isolate in our GFEL plate #46 at the position of row E column 5, inhibited *P. falciparum* binding to midgut FREP1 with an inhibition rate of 99%. In vivo, standard membrane feeding assays (SMFA) found that mosquito midguts in the control group had a significant number of oocysts (red dots in Figure 1a). Out of 33 mosquitoes in the control group, 93% mosquitoes possessed oocysts (Figure 1c). Notably, 100 µg/mL GFEL-46E5 rendered a mosquito midgut free of oocysts (Figure 1b). Out of 32 mosquitoes fed with *P. falciparum*-infected blood containing GFEL-46E5, only six mosquitoes had oocysts (Figure 1c). The mean oocysts per midgut were reduced by 99% in the mosquitoes fed the GFEL-46E5 fungal extract compared to that of the control group (Figure 1c).

### 2.2. Identification of the Fungal Species

Next, we identified the fungal species of the candidate through its morphology and genomic sequence. The fungal colony proliferated on the malt extract agar (MEA) medium. After 3–5 days, the front side of the colony showed a purplish-black color (Figure 2a). The backside of the colony was yellow (Figure 2b). The fungus has back conidial heads under the microscope, which are spherical or radiate, splitting into well-defined divergent columns (Figure 2c). Furthermore, its conidiophores have smooth-walled stipes, are hyaline, or are slightly pigmented at the apex (Figure 2c). The conidia are hyaline to brown, conspicuously echinulate, and sub-spherical (Figure 2d) to ellipsoidal (Figure 2e).

We also sequenced its internal transcribed spacer region (ITS) to identify its species. We extracted the fungal genomic DNA, PCR-amplified the ITS DNA fragment using the fungal genomic DNA as templates and sequenced the PCR product. The sequence was 99.8% identical to *Aspergillus aculeatus* isolate 3 ITS (accession NR_111412) sequence in the NCBI database (Figure 2f). Only one base pair (at position 276) changes from G to A. We concluded that the GFEL-46E5 fungus isolate belongs to *A. aculeatus* species according to its morphology and ITS sequence.

### 2.3. Isolation of Active Compounds

The *A. aculeatus* GFEL-46E5 isolate was fermented for four weeks in mushroom bags to obtain enough materials for isolation and exploration of transmission-blocking activity. The cultured fungus extracted with ethyl acetate was subjected to flash column chromatography, and the column was eluted with hexane to remove nonpolar compounds. Then, the column was eluted sequentially with 100% hexane, 50% ethyl acetate in hexane, 5% methanol in dichloromethane, and 100% methanol sequentially to obtain fraction 1, 2, 3, and 4, respectively (Figure 3a).

These four fractions were dried and used to examine their activity in inhibiting *P. falciparum* transmission to mosquitoes using SMFA. The results of the infection assays showed that fraction 3 was able to completely inhibit parasite infection in mosquitoes at ten µg/mL, fraction 4 partially inhibited malaria infection in mosquitoes, and fraction 1 and 2 did not inhibit parasite infection in mosquitoes (Figure 3b). Therefore, we focused on fraction 3 to isolate the pure active compounds.

In thin layer column chromatography, a significant component has an Rf value of 0.55 using the solvents of methanol and dichloromethane in the ratio of 1:10, and the primary compound in fraction 3 is a yellowish oily liquid and accounted for 80% of the total dry weight. Using LC-MS spectra, the major peak of fraction 3 is at the retention time of 12.9 m (Figure 4a) and showed the exact *m/z* of 237.1466, 219.1360, 173.1306 (Figure 4b). The *m/z* value of 237.1466 represented the form of [M+H]^+^, 219.1360 represented the form of [M+H-H_2_O]^+^, and 173.1306 represented the form of [M+H-2×H_2_O-CO]^+^.

Fraction 3 was applied to semi-preparative HPLC to obtain the pure active compound (Appendix A). Only one peak appeared (Figure 5a), and its UV absorbance profile showed a maximum absorbance at 242 nm and a minimum absorbance at 217nm (Figure 5b). This pure compound is referred to as “C1”. SMFA determined the transmission-blocking activity of C1 and examined a serial dilution with the final concentrations of 0, 4, 8, or 40 µM. The results show that 8 µM of the compound was able to significantly inhibit *P. falciparum* transmission (*p* < 0.009) and 40 µM compound was able to inhibit parasite infection in mosquitoes completely (Figure 5c). The average numbers of oocysts in mosquitoes treated with 0, 4, 8, or 40 µM of C1 were 4.8, 4, 0.09, and 0.06, respectively. The infection prevalence was 67%, 74%, 40% and 6.3%, respectively. The IC_50_, defined as the concentration of C1 that reduces 50% infection prevalence rate in mosquitoes compared with that of the DMSO control, was calculated as 7.89 µM.

### 2.4. Identification of the Pure Active Compound

The high-resolution mass spectrometry of C1 was analyzed. It detected the exact mass of the pure compound of [M+H]^+^ peak at 237.1481 (Figure 6a), which is in agreement with the mass spectrometry data of a previously reported compound asperaculane B (1, Figure 6b) isolated from the same species of fungus [8]. The calculated mass of asperaculane B with a proton is 237.1485 [12]. Its formula is C_14_H_21_O_3_. Therefore, we hypothesize that C1 is asperaculane B.

Instead of repeating previously published work [12,14,15], we obtained a set of pure compounds (Figure 6b), including the pure asperaculane B (Appendix A) from two laboratories, where the compounds were isolated from *A. aculeatus* in 2015 [12] and 2019 [14]. We determined the malaria transmission-blocking activity of these compounds with SMFA at 10 µg/mL and fed *An. gambiae*. The result showed that asperaculane B was able to inhibit *P. falciparum* transmission to mosquitoes while the other three, including aculene A, aculene D, and secalonic acid D, did not (Figure 6c). Moreover, SMFA measured the different concentrations of asperaculane B on *falciparum* transmission and found that asperaculane B significantly inhibited *P. falciparum* transmission at 10 µM (*p* = 0.003) and 40 µM (Figure 6d). Its IC_50_ was calculated as 8.48 µM and it was very close to the IC_50_ (7.89 µM) of C1.

### 2.5. Asperaculane B Inhibits Asexual Stage P. Falciparum

We also determined the inhibition efficacy of *asperaculane* B on asexual stage *Plasmodium* development. One µL *asperaculane* B in DMSO with different concentrations was added into 150 µL *P. falciparum* culture, and the parasitemia was monitored in the following three days. During this process, we replaced the medium on day two, and added the same amount of asperaculane B. Results (Figure 7) show that at low concentrations, e.g., <0.02 μg/mL, the inhibition is not significant. However, the average parasitemia in replicates at 0.2 μg/mL and 1 μg/mL asperaculane B concentration was 0.7% and 0.3% on day 3 respectively, about 30% and 10% of control, respectively. Collectively, these data indicate that asperaculane B can also inhibit asexual stage parasite development in blood. The IC_50_ on day three was 0.72 μg/mL or 3 μM.

### 2.6. Cytotoxic Effects of Asperaculane B

Finally, we analyzed the cytotoxicity of asperaculane B as an initial step to understand its molecular mechanism and develop a drug. Asperaculane B was added into human embryonic kidney 293 (HEK293) cell culture. After incubation for 24 h, MTT (3-(4,5-dimethylthiazol-2-yl)-2,5-diphenyltetrazolium bromide) assay showed no significant cytotoxic effects of this compound at the concentrations of ≤120 μM (Figure 8). A substantial reduction in cellular growth was detected only when the level increased to 400 μM. Notably, the 40 μM compound was able to inhibit malaria transmission in mosquitoes completely.

### 2.7. Asperaculane B Prevents FREP1 from Binding to P. Falciparum-Infected Cell Lysate

Since the candidate fungal extract was obtained by screening the GFEL for their inhibition on FREP1-*falciparum* interaction, we determined whether asperaculane B could prevent FREP1 from binding to *P. falciparum*-infected cell lysate. Microplates were coated with *P. falciparum*- infected cell lysates. Then asperaculane B was mixed with FREP1 protein to make final concentrations of 10, 20, 40, and 80 μM, which were loaded to the microplate. The DMSO (1%, *v/v*) was a non-inhibition control. The results showed that asperaculane B significantly inhibited the interaction between FREP1 protein and *P. falciparum*-infected cell lysate, and the inhibition was also dose-dependent (Figure 9).

## 3. Discussion

The international malaria communities are challenged by the fast spread of insecticide-resistant mosquitoes and drug-resistant *Plasmodium* parasites, as well as difficulties in vaccine development. It makes the control even more difficult since a patient under the treatment of conventional antimalarial drugs can still transmit malaria before he/she completely recovers. Therefore, novel drugs that combine the treatment of malaria patients and malaria transmission prevention are essential for malaria control. 

*Plasmodium falciparum* growth assays are used widely to screen drugs against sexual stage malaria, e.g., traditional antimalaria drugs to control blood-stage parasites [7]. It is urgently needed to identify potential drugs that cut the malaria transmission cycles. The newly discovered FREP1-mediated *Plasmodium* transmission pathway [10] provided us with an ideal target to block malaria transmission. Since FREP1 plays its function through binding to parasite-infected cells, screening small molecules through ELISA that inhibit the interaction is an optimal high throughput assay [8]. We demonstrated this approach by screening our newly constructed global fungal extract library and discovered an *A. aculeatus* isolate, GFEL-46E5, that effectively inhibited *P. falciparum* transmission to mosquitoes.

*A. aculeatus* belongs to an *Aspergillus* fungal subgroup, section *Nigri* (the black aspergilli), which consists of 18 species [16]. Several drug candidates against tumors or bacteria, e.g., okaramines H and I, have been isolated from this group [15]. However, some of *A. aculeatus* species also release mycotoxins [17,18]. The *P*-orlandin from *Aspergillus niger*, another member of this section, has been isolated, and it significantly inhibits *P. falciparum* transmission to mosquitoes [8].

The active compound from *A. aculeatus* strain GFEL-46E5 was identified as asperaculane B, a sesquiterpenoid. This paper is the first to report that asperaculane B has antimalarial activity. Notably, asperaculane B inhibited both the sexual stage and asexual stage *P. falciparum*, which should benefit public health communities in controlling malaria substantially. Asperaculane B inhibits the interaction between FREP1 and parasites, which may be responsible for its activity of blocking malaria transmission. The molecular mechanism of asperaculane B against blood-stage malaria is unknown. Asperaculane B has been shown to have nontoxicity to human cancer cells and regular hamster cell lines even at the concentration of 50 µM [12], much higher than its IC_50_ against malaria parasites. Here, we demonstrated that asperaculane B had no cytotoxic effects on HEK293 cells at the concentrations of 120 μM. Therefore, the impact of asperaculane B against sexual and asexual *P. falciparum* is not through general cytotoxicity. 

As a natural product, asperaculane B has some advantages for future drug development. Firstly, *Aspergillus* fungi have been used as a resource for drug discovery. For example, aspergillusol A is an α-glucosidase inhibitor and works as an oral anti-diabetic drug [19], and aculeacins A–G have antifungal and antibiotic activities [20]. Therefore, many features are known, and related technologies are ready. Secondly, a large amount of fungal secondary metabolites can be obtained by large-scale fermentation. Moreover, the recent development of genomic sequencing technology and the identification of more biosynthetic gene clusters speed up the process of mass production of the active fungal metabolites and significantly reduce the cost of mass production of this compound. Thirdly, this compound did not show toxicity to human and mammalian cells [12], and it is not a mycotoxin. Finally, its derivatives or other druggable features, and the accumulated experience of total synthesis of sesquiterpenoid compounds from academia and industry should help accelerate the process of drug development [21]. Collectively, these features make asperaculane B an excellent antimalarial and transmission-blocking drug candidate for future development.

## 4. Materials and Methods

### 4.1. Maintenance of An. Gambiae Mosquitoes

The mosquitoes (*An. gambiae* G3 strain) were maintained in an insectary room maintained at 27 °C, 80% humidity with a 12-h day/night cycle. Adults were fed on 8% sucrose and fed with mouse blood for egg production, and larvae were fed daily with 0.1 mg grounded fish food.

### 4.2. Screening a Fungal Extract Library with an ELISA Method

Our research group recently constructed GFEL. GFEL contains about 10,000 isolates from soil or plants collected worldwide. Each fungal species was grown on cereal-based medium at 25 ± 2 °C for four weeks to produce metabolites. The ethyl acetate (Thermo Fisher Scientific, Waltham, MA, USA) was used to extract small molecules, and the dry extracts were dissolved in dimethyl sulfoxide (DMSO) (Thermo Fisher Scientific, Waltham, MA, USA) at the concentration of 10 mg/mL stock. In vitro, high throughput screening was based on the ELISA method described in detail previously [8]. First, we coated 100 μL 15-day old cultured *P. falciparum*-infected cell lysates (2mg/mL proteins in PBS containing 0.2% Tween-20) for one hour at RT (room temperature). One microliter of each crude extract with a concentration of 2 mg/mL in DMSO was mixed with 99 μL FREP1 proteins in PBS (1:99) to determine whether an extract can disrupt the interaction of FREP1 and *P. falciparum* lysate. The fungal extracts that reduced the ELISA signals (A_405_) of FREP1 binding to *P. falciparum*-infected red blood cell lysates were considered to contain active compounds. The mixture containing 1% DMSO was a negative control.

### 4.3. Mass Production and Extraction of Fungal Metabolites

The fungal isolates were cultured in liquid malt extract medium (MEA) (Life Tech, Grand Island, NY, USA) for 3-5 days and transferred to one autoclaved mushroom bags containing 550mL of sterile and dried Cheerios cereal (General Mills, Minneapolis, MN, USA) and 980 mL of 0.3% sucrose supplemented with 50 mg/L chloramphenicol (Life Tech, Grand Island, NY, USA) at 25 ± 2 °C for four weeks. The fungal metabolites were extracted with one litter of ethyl acetate twice, and the organic layer was filtered with the two layers of cheesecloth in the Buchner funnel. The resultant ethyl acetate supernatant was dried using a rotary evaporator (Heidolph, Elk Grove Village, IL, USA).

### 4.4. Transmission-Blocking Assays

*P. falciparum* was maintained with RPMI-1640 medium (Life Tech, Grand Island, NY) containing 10% heat-inactivated human AB+ serum (Interstate blood bank, Memphis, TN), 12.5 μg/mL hypoxanthine and 4% hematocrit (O+ human blood) in a candle jar at 37 °C for 15–17 days. The medium was replaced every day. Parasitemia or gametocytaemia were monitored every other day under a light microscope using Giemsa-stained blood smear.

The 15–17-day old *P. falciparum* cultures containing 2–3% gametocytes at stage V were collected and diluted with new O+ type human blood, and mixed with the same volume of heat-inactivated AB+ human serum. Then, the candidate fungal extract, fractions, or pure compounds in DMSO were mixed with the infected blood (the final DMSO concentration in blood was <1%). SMFA was performed to feed ~100 3–5 days old female mosquitoes for 30 m. The midguts were dissected seven days post-infection and stained with 0.1% mercury dibromofluresein disodium salt (Sigma-Aldrich, St. Louis, MO, USA) in PBS. The oocysts were counted under light microscopy.

### 4.5. Determination of Fungal Species

The fungal isolate was grown in liquid MEA at 25 ± 2 °C for three days, and the collected fungi were used for DNA extraction. The genomic DNA applied as a PCR template was isolated using DNAzol Reagent from the collected fungi following the manual (Life Tech). Next, the nuclear ribosomal internal transcribed spacer (ITS) region was amplified with the primer set of ITS1F (5′-CTTGGTCATTTAGAGGAAGTAA-3′) and ITS4 (5′-TCCTCCGCTTATTGATATGC-3′) primers and Phusion DNA polymerase (Thermo Fisher Scientific) with the method as the following: 98 °C 30 s; 98 °C 15 s, 55 °C 15 s, 72 °C 15 s, 35 cycles; 72 °C 5 m. The amplified products were sequenced, and the sequences were blasted against the NCBI database for the identification of the fungal species.

### 4.6. Purification of the Active Compounds by Gravity Chromatography and High-Performance Liquid Chromatography

The solvents used for extraction and separation were all from Thermo Fisher Scientific if not explicitly mentioned. The crude extract (1 g) was loaded into a column with the diameter and length of the column of 2.5 cm and 30 cm, respectively. The fine silica used for column chromatography was of particle size 40–63 µm (60 × 100 mesh) (Sigma Aldrich, St. Louis, MO, USA). Then, the column was eluted with 200 mL of hexane, followed by elution sequentially with 400 mL of 50% ethyl acetate in hexane, 500 mL of 5% methanol in dichloromethane and 300 mL of 100% methanol. All four fractions were dried, and their activities in blocking malaria transmission were determined as described above. The fractions containing active compounds against malaria transmission were further fractioned using a Shimadzu HPLC system (an LC-20AD pump, SPD-M20A UV and visible detector, and FRC-10A fraction collector, Shimadzu, Columbia, MD, USA). A semi-purification column (Gemini C18 250 mm × 10 mm, 5 μm, Phenomenex, Torrance, CA, USA) and a gradient solvent of MeOH-H2O (50:50–100:0) was used to obtain the pure bioactive compound.

### 4.7. LC-MS and High-Resolution Mass Spectrum

LC-MS includes Agilent 1100 series LC system consisting of G1312B binary pump, G1313A autosampler and 1200 Series diode-array detector (Agilent Technologies, Santa Clara, CA, USA) with an analytic column (Hypersil GOLD aQ, 3 µm C18, 150 x 2.1 mm) (Thermo Fisher Scientific) run on a gradient solvent of MeOH-H2O (50:50–100:0) at a flow rate of 0.2 mL/m. The eluted compounds were then analyzed with mass spectrometry using (+) ESI mode on the Agilent 6220 TOF mass spectrometer (Gas temperature - 350˚C, Drying gas (N2)–8 L/m. Nebulizer–2 Bar).

High-resolution mass spectrometry was recorded using (+) ESI mode on the Bruker Daltonics, Impact II QTOF mass spectrometer (Gas temperature −200 °C, Drying gas (N_2_)–4 L/m Nebulizer–0.3 Bar, Bruker Scientific LLC, Billerica, MA, USA). Both LC-MS and high-resolution mass spectrometry analyses were conducted at the University of Florida Mass Spectrometry Research and Education Center, Department of Chemistry.

### 4.8. Asexual Plasmodium Falciparum Growth Inhibition Assays

*P. falciparum*-infected human red blood cells were mixed with new AB+ type uninfected human RBCs to prepare cultures with 0.5% parasitemia and 2% hematocrit. The compound was dissolved in DMSO at the concentration of 1mg/mL and diluted to various concentrations with DMSO. About 2 μL of small molecule solution was added into 1 mL of culture per well in a 24-well plate. The plate was incubated in a candle jar at 37 °C. The medium, as well as the candidate compound, was replaced at 48 h. The parasitemia was recorded at 24 h, 48 h, 72 h, and 96 h after incubation. IC_50_ was determined by analysis of the dose-response curve made by GraphPad Prism (GraphPad Software, CA, USA).

### 4.9. Cytotoxic Assay

The MTT (3-(4,5-dimethylthiazol-2-yl)-2,5-diphenyl tetrazolium bromide) cell proliferation assay (Thermo Fisher Scientific) was used to measure the general cytotoxicity of pure compounds to human embryonic kidney 293 (HEK293) cell line. The HEK293 cells at a concentration of 2 × 10^5^ cells/well in 100 μL culture medium containing different levels of the pure compounds, e.g., 10–400 µM)] were seeded into microplates and incubated for 24 h at 37 °C and 5% CO2. Next, the cells were incubated with 100µL of MTT reagent for four h at 37°C (final concentration 0.5 mg/mL). After removing all but 25 µL of the medium from the wells, 100 µL of DMSO was added to each well and incubated for 10 m to dissolve the purple formazan crystal for measurement. The optical density at an absorbance wavelength of 540 nm was measured with the plate reader.

### 4.10. Statistical Analysis

All the experiments were independently repeated at least twice. As reported previously [8,9,10], the number of oocysts in mosquito midguts does not follow normal distribution. Therefore, nonparametric statistical analysis, the Wilcoxon–Mann–Whitney test, were used to calculate the *p*-value of infection difference between control and experimental groups in each assay with Prism (GraphPad Software, CA, USA).

## Figures and Tables

**Figure 1 molecules-25-03018-f001:**
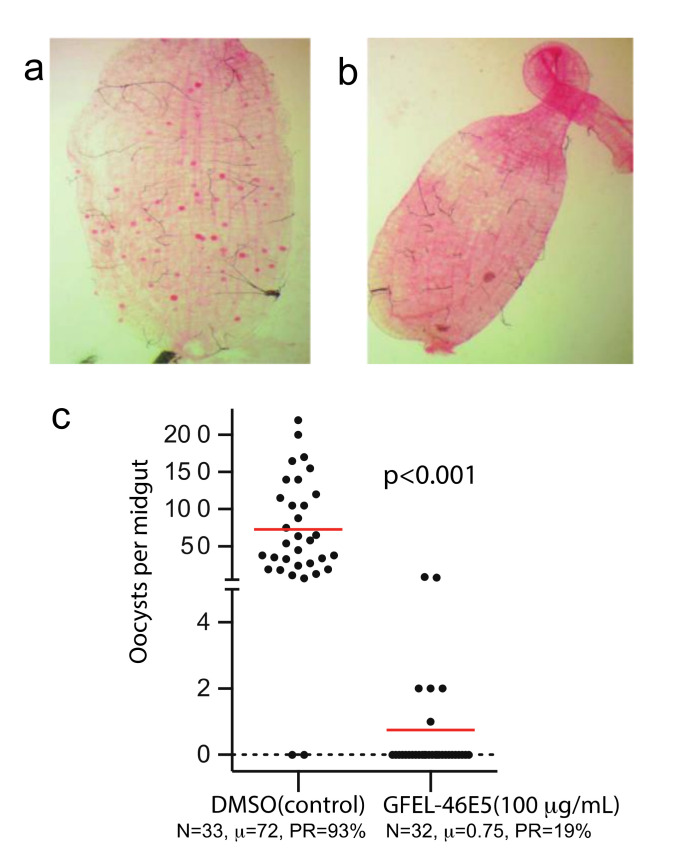
The candidate fungal extract (GFEL-46E5) significantly inhibited *Plasmodium falciparum* infection in mosquito midguts. (**a**) In the control group, a mosquito midgut was infected with *P. falciparum*. Each red dot represents one oocyst. (**b**) A mosquito midgut from GFEL-46E5 treated group did not possess *P. falciparum* oocysts. (**c**) Mosquitoes fed with *P. falciparum* containing GFEL-46E5 extract or dimethyl sulfoxide (DMSO, control) showed different numbers of oocysts. Each spot represents the number of oocysts in a mosquito midgut. The difference between the two groups was significant (p < 0.001). N: # of mosquitoes for each treatment; μ: the mean oocysts per midgut; PR: infection prevalence in mosquitoes. The assays were repeated independently three times.

**Figure 2 molecules-25-03018-f002:**
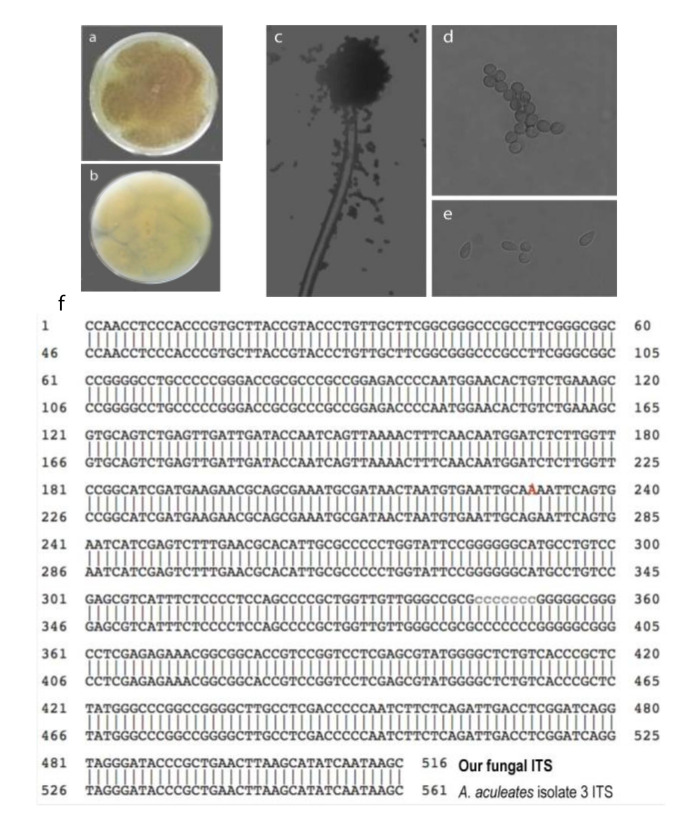
The candidate fungus GFEL-46E5 is an *Aspergillus aculeatus* isolate according to morphology and ITS (internal transcribed spacer) sequence. (**a**) Top view of the fungus colony on MEA; (**b**) Bottom view of the fungus on MEA; (**c**) conidial head; (**d**) subspherical conidia; (**e**) ellipsoidal conidia; and (**f**) ITS sequence aligns with the ITS of *Aspergillus aculeatus* isolate three from NCBI (Access #: NR_111412).

**Figure 3 molecules-25-03018-f003:**
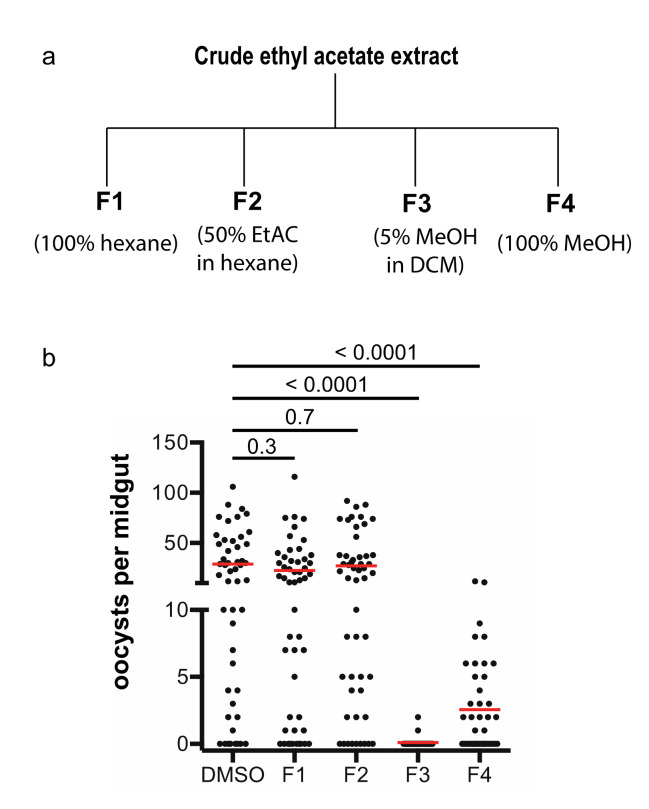
Fractions and activity analysis. (**a**) The crude extract was fractioned by a gravity silica column. (**b**) The activities of four fractions in inhibiting malaria transmission were determined by SMFA (standard membrane feeding assays).

**Figure 4 molecules-25-03018-f004:**
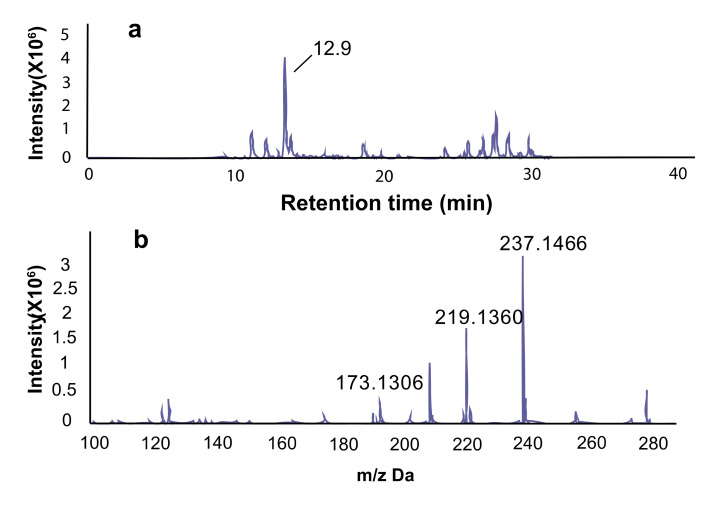
Isolation and purification of active compounds. (**a**) The active fraction 3 was subjected to LC-MS. The significant component is at 12.9 m in the LC; (**b**) the exact *m/z* from the significant component was 237.1466 ([M+H]^+^, C_14_H_21_O_3_; calculated 237.1485), 219.1360 [M+H-H_2_O]^+^, 173.1306 [M+H-2×H_2_O-CO]^+^.

**Figure 5 molecules-25-03018-f005:**
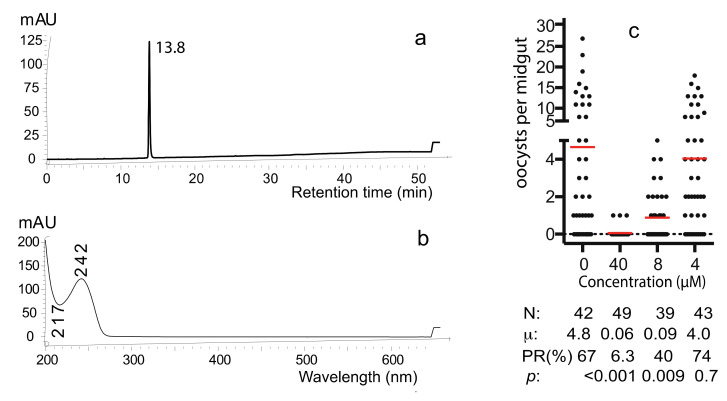
The characteristic profiles of the pure active compound. (**a**) HPLC profile showed one peak. (**b**) UV-Vis absorbance profile. (**c**) The pure compound inhibited *P. falciparum* transmission to *An. gambiae* in a dose-dependent manner. N: # of mosquitoes for each treatment; μ: the mean oocysts in a midgut; PR: infection prevalence in mosquitoes; *p*: statistical values compared to the control (no compound) by Wilcoxon test.

**Figure 6 molecules-25-03018-f006:**
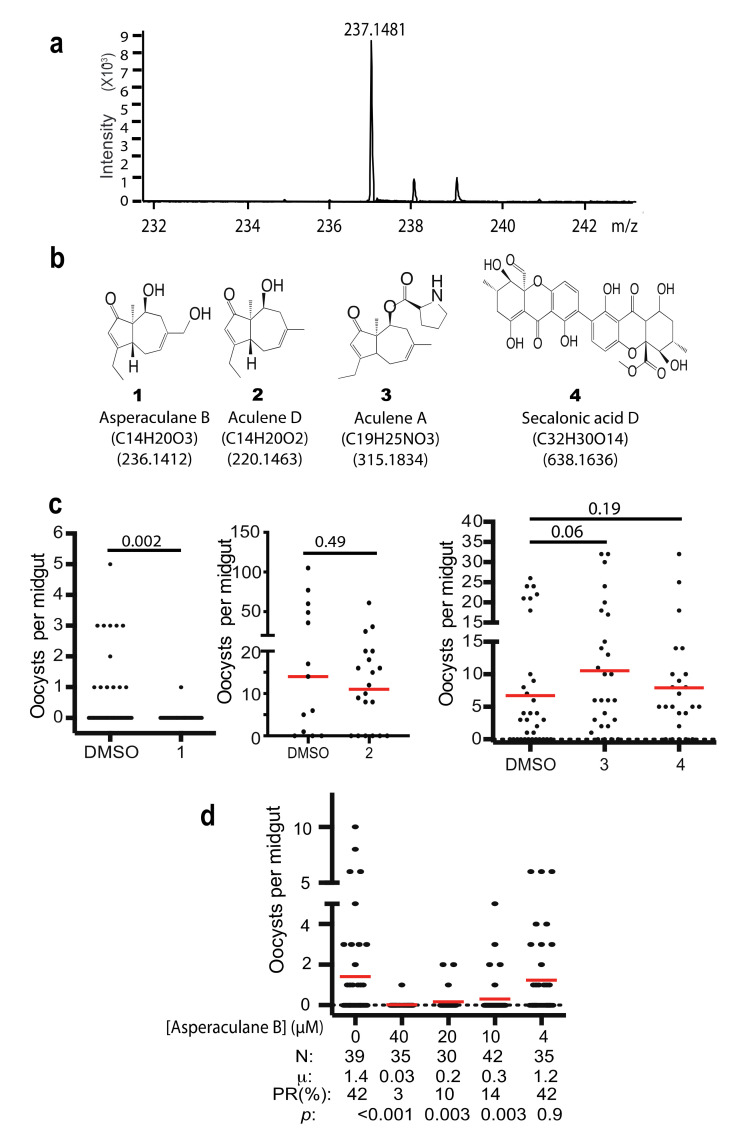
Identification of the active compound. (**a**) The high-resolution mass spectrometry of the HPLC purified compound with a proton (236.1403 + 1.0078 = 237.1481). (**b**) Four purified and identified compounds from *A. aculeatus* from other research labs [12,14]. (**c**) The transmission-blocking activities of four compounds. (**d**) The transmission-blocking activity of serial dilution of asperaculane B. N: # of mosquitoes for each treatment; μ: the mean oocysts per midgut; PR: infection prevalence in mosquitoes; *p*: The statistical *p* values of each concentration comparing to the control (concentration was 0).

**Figure 7 molecules-25-03018-f007:**
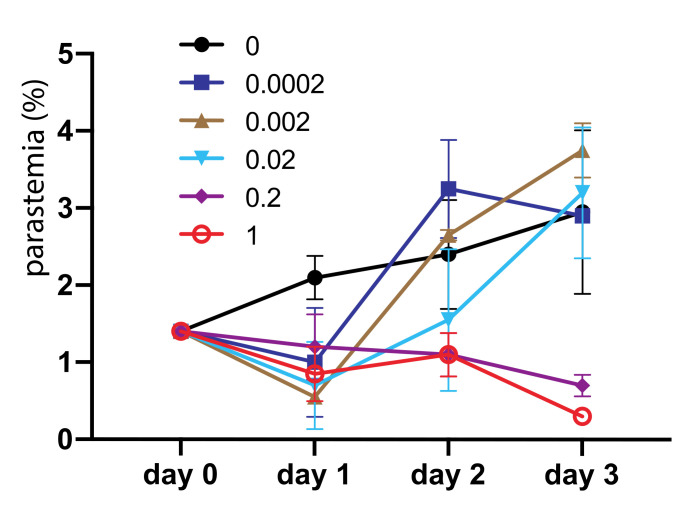
Asperaculane B inhibits the development of asexual stage *P. falciparum*, and the inhibition is dose-dependent.

**Figure 8 molecules-25-03018-f008:**
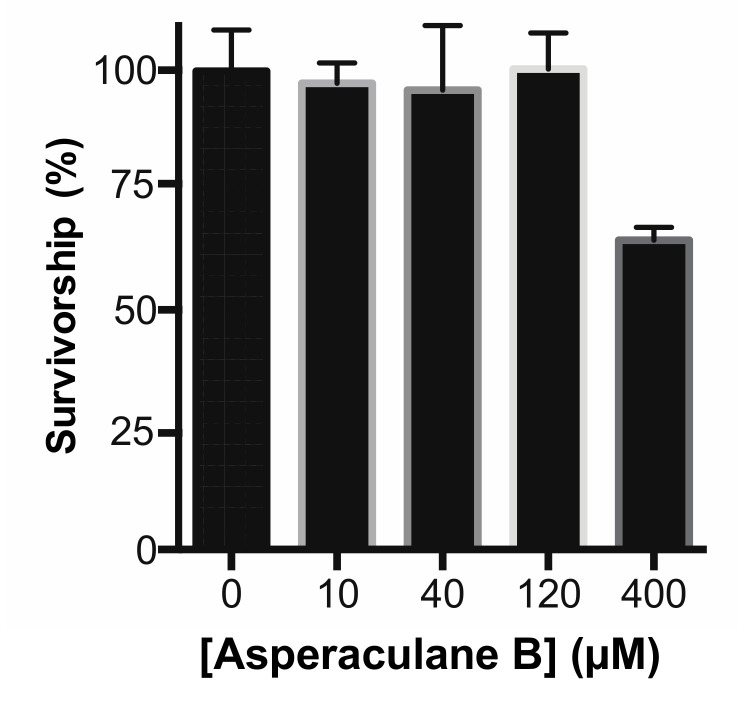
Cytotoxicity of asperaculane B at the varying concentrations (0-400 µM) on the human kidney cell line (HEK293).

**Figure 9 molecules-25-03018-f009:**
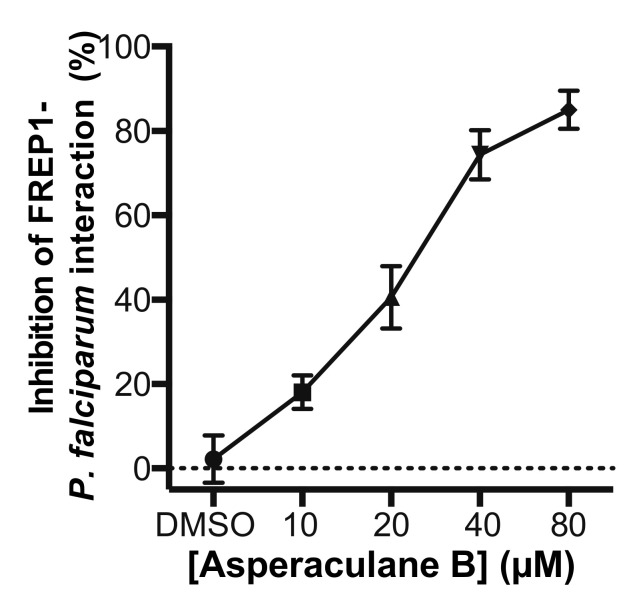
Asperaculane B prevented FREP1 proteins from binding to *P. falciparum*-infected cell lysate determined by ELISA.

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
