# Peer review of "Fungal Metabolite Asperaculane B Inhibits Malaria Infection and Transmission"

_molecules, 2020, doi:10.3390/molecules25133018_

Round 1

Reviewer 1 Report

Niu et al. describe that asperaculane B inhibits malaria infection and transmission. This manuscript is well written and fits the scope of the journal Molecules. Since there are a few mistakes, please improve them in their revise process.

- L65 Please spell “SMFA” out.

- L70-72 Please delete the sentences in the template.

- Fig. 3 “Crude ethyl acetic extract” -> “Crude ethyl acetate extract”

Author Response

We thank the reviewer for valuable comments. The concerns from this reviewer have been addressed as the following:

  • L65 Please spell “SMFA” out.

* Done. Line 69

  • L70-72 Please delete the sentences in the template.

* Done.

  • Fig. 3 “Crude ethyl acetic extract” -> “Crude ethyl acetate extract”

* Corrected in Fig 3. 

Reviewer 2 Report

The authors describe the chemical purification and biological evaluation of asperaculane B. Its antimalarial activity was determined, showing its effect by blocking infection and transmission. As demonstrated in the manuscript, the authors have experience working with fungal extract libraries.

This manuscript is not suitable for publication in its current state. Some changes and clarifications should be carried out before being considered for publication.

Here are some comments that I would like to be addressed by the authors.

  • Related to the introduction section

I think that the introduction should be more transparent with the public and indicate that the group has previously managed to isolate and biologically evaluate this type of molecule or even the same molecule. The reader just found out about that in the results part, and that shouldn't be the case. In the introduction, the articles that should go are cited, however where the molecule itself is mentioned as previous work is in the results (2.4 Identification of the pure compound, line 146, page 6). I consider that perhaps it is a strategy of the authors to be able to show the newness of the manuscript (very valid); however, the authors should not forget that this information is state of the art and should be clearly mentioned in the introduction.

  • Related to Identification of the pure active compound

One of the main parameters when checking the biological activity of a new substance is purity. The last mentioned is because the reported biological activity may be due to an impurity present in a chemical sample. No direct purity assays such as proton or carbon nuclear magnetic resonance are reported in the manuscript, even an image of a sample single layer chromatography would help to verify this. Although it is true, mass spectroscopy tests seem to indicate that if it has the substance, it is not commonly used to determine purity. I would ask the authors to ATTACH the necessary assays for the determination of purity and the correct chemical elucidation of the structure. Even the authors could include an image in supplementary material where you can see a thin layer chromatography of the isolated substance compared to the total mixture. A very typical image in any workbook when conducting a bio guide of natural products

  • From the statistical point of view

The authors mention that they performed Wilcoxon-Mann-Whitney test, which is reserved for non-parametric type tests. However, they work with the mean of the values ​​as a summary measure of the numerical data, which is reserved only for parametric tests. Even more severe, the measure of the dispersion of the data should be the standard deviation (SD) (if working with the mean). Still, the SD is absent throughout the manuscript for its biological activity values ​​mainly, even though they report that they were performed twice. Please, I would ask the authors to solve this problem because there could be a mishandling of data.

Also, there are some things that I think are not being mentioned in the statistical part. For example, in figure 3 and 6 where the different tests performed are shown, from the report of the p-values ​​it seems that they also used other statistical tests such as Kruskall Wallis (if it is true that they used non-parametric tests, otherwise it would be ANOVA)

Author Response

Thank the constructive comments from this reviewer. We address all comments as following:

  • Related to the introduction section

Answer: the manuscript was written the same order as how we designed and conducted the research. We screened a fungal library and identified a candidate fungus. Then we isolated active compound. When we tried to determine the structure of the active compound, we read literatures and found our collaborative labs. In the revision, we introduce the fungus more and move some citations from results to the introduction (line 57-60): Many bioactive secondary metabolites have been isolated from this fungal species [12, 13]. The active compound was isolated and determined to be asperaculane B. The structure of asperaculane B has resolved previously [12, 14].

  • Related to Identification of the pure active compound

Answer: As request, we added TLC data as Supplemental Figure S1 (line 144, 169) to demonstrate the purity of the compound. Single band was shown from The isolated active compound showed one band and had the same Rf as the reference compound. 

  • From the statistical point of view

Answer: The number of oocysts in mosquito midguts does not follow normal distribution. Parametric test cannot be used. The nonparametric tests were used to calculate the p-value of infection difference between control and experimental group. We added this justification at section 4.10, line 372-374.